# Self-Compatibility Not Associated with Morphological or Genetic Diversity Reduction in Oil-Rewarding *Calceolaria* Species

**DOI:** 10.3390/plants9101377

**Published:** 2020-10-16

**Authors:** Maureen Murúa, Anahí Espíndola, Fernanda Pérez

**Affiliations:** 1Centro GEMA, Genómica, Ecología y Medio Ambiente, Facultad de Estudios Interdisciplinarios, Universidad Mayor, Camino La Pirámide 5750, Santiago 8580745, Chile; 2Departamento de Ecología, Facultad de Ciencias Biológicas, Pontificia Universidad Católica de Chile, Casilla 114-D, Santiago 8331150, Chile; mperezt@bio.puc.cl; 3Department of Entomology, Plant Sciences Building 3138, University of Maryland, College Park, MD 20742-4454, USA; anahiesp@umd.edu

**Keywords:** *Calceolaria*, self-incompatibility, Chile, specialized pollination, floral morphology, plant reproduction

## Abstract

One of the most common evolutionary transitions in angiosperms is the reproductive change from outcrossing to selfing, commonly associated with changes in floral biology and genetic diversity. Here, we aim to test whether self-compatibility leads to a reduction of floral traits and genetic diversity. For this, we experimentally estimate levels of self-compatibility, measure three floral traits and estimate four genetic diversity parameters using nine microsatellites in nine *Calceolaria* species. Our analysis indicated that four of the study species were self-incompatible. In addition, we found that self-compatible species did not show a reduction in floral traits size, but rather displayed larger corolla and elaiophore areas. Our analyses of genetic diversity identified larger allele number and observed heterozygosity in selfers than in outcrossers, but did not find larger inbreeding in the self-compatible species. Even though our results contradict our expectations, in the case of *Calceolaria*, their high dependence on only two genera of oil-bees puts the genus in a vulnerable reproductive position, probably facilitating the evolution of reproductive assurance mechanisms in the absence of pollinators. As a result, plants maintain their pollinator attraction traits while evolving the ability to self, possibly in a delayed way.

## 1. Introduction

One of the most common evolutionary transitions in angiosperms is the reproductive change from outcrossing to self-fertilization, which has occurred independently in many lineages [1,2,3]. This transition has been associated with changes in floral biology, ecology and genetics, with selfers showing reduced floral display and hercogamy index, rapid plant growth, and increased inbreeding depression [4,5]. Resource allocation theory has frequently been invoked to explain these changes, stating that once the transition to selfing occurs, the energy used in building large flowers can be reallocated to other functions, such as ovule production [6,7]. However, this reallocation is also dependent on pollination mode (e.g., animal, wind), since if animal pollinators are needed for pollen transfer (i.e., selfing is not autonomous), some of the floral attraction features will be maintained [8]. The hercogamy index has also been shown to be reduced in plants with higher abilities of selfing [9], and is explained by the fact that reducing the distance between the reproductive organs improves the ability of the plant to self-pollinate in the absence of pollen carriers. Finally, the genetic diversity of the species is also expected to vary in selfers vs. outcrossers. Indeed, the former are expected to display lower genetic diversity than the latter, at least at the population level [10]. Taken together, one can expect that the negative consequences of selfing, both morphological and genetic, will be greater in plants where fertilization occurs automatically than in those where it requires cross-pollination [11].

Pollination specialization is expected to promote greater precision in floral handling, improving reliable pollen removal and deposition, reducing pollen limitation, and increasing outcrossing [12]. However, in response to the variability of the pollinator environment, plant species with specialized pollination systems can sometimes evolve autonomous selfing [12,13,14]. Accordingly, two modes of selfing may be favored: (i) early selfing, which takes place before the arrival of outcrossing pollen and should be favored when pollinators are absent or flower density is low; and (ii) delayed selfing, which occurs at the end of a flower’s life span and is favored when pollinator visitation is variable or unpredictable [15]. Among these two, the latter is considered more advantageous, because it does not decrease outcrossing potential and incurs no pollen or seed discounting [15]. In such cases, it has been observed that floral traits involved in outcrossing are retained (e.g., large flowers, large number of flowers, floral cues, pollinator rewards [8]), and it has been proposed that the costs of maintaining them are outweighed by the genetic benefits of an eventual cross-pollination [8]. Even though changes in trait values have been studied many times, the genetic consequences of the transition from outcrossing to selfing in plants displaying specialized pollination are still little understood.

The nectarless oil-rewarding plant genus *Calceolaria* L. and the oil-bees genera *Centris* Fabricius and *Chalepogenus* Holmberg interact to form a fascinating specialized American plant-pollinator system. In this system, *Calceolaria* flowers harbor an oil-secreting gland (the elaiophore), from which the oil-bees collect floral oils while passively pollinating the flowers [16,17,18,19]. Although the majority of the plants in the system are supposed to require insects for reproduction, investigations have shown that some species are also capable of low-level autonomous selfing [16,20]. This has been hypothesized to evolve in response to a lack of pollinators and/or the unreliability of the pollination environment [16]. However, the extents of the transition between strictly outcrossing to selfing, and its morphological and genetic consequences have never been tested in this group.

In the present work, we propose to test the idea of reduced floral attraction traits and genetic diversity in self-compatible species using manipulative pollination experiments and microsatellites on nine *Calceolaria* species belonging to the subgenus *Cheiloncos* (Figure 1a). Specifically, we aim to address the following questions: (a) what is the breeding system of the nine studied species? (b) is there a reduction in the floral traits associated with pollinator interaction (i.e., corolla area, hercogamy index, elaiophore area; Figure 1b) in self-compatible vs. self-incompatible species? and (c) do self-compatible species have lower genetic diversity than outcrossers? To do this, we first experimentally estimated the levels of incompatibility using pollination crosses in field conditions, then measured three floral traits important to the plant-pollinator interaction, and finally estimated four genetic diversity parameters using data from nine polymorphic microsatellites. Our expectations were that self-compatible species would display reduced pollinator attraction traits and genetic diversity.

## 2. Materials and Methods

### 2.1. Species and Breeding System Characterization

In order to determine the breeding system of nine *Calceolaria* species, a hand-pollination experiment was performed in field conditions (Figure 1). During the summer seasons of 2016–2018, thirty to fifty plants per species were selected and tagged in eight different localities throughout central Chile (Table 1). Each population was located in unprotected areas for which no collection permits are required, and separated by at least 1 km. Species identification was done in situ following the review and description of Chilean species by Ehrhart [17] and no herbarium material was collected.

In the flowering peak, six buds per plant were chosen, covered with a tulle mesh and two buds per plant were randomly assigned to three different treatments (i.e., 6 buds = 2 buds × 3 treatments): (a) automatic self-fertilization (AS): floral bud marked and without manipulation, (b) geitonogamy (G): emasculated bud manually pollinated with pollen of the same plant, and (c) xenogamy (X): emasculated bud pollinated by manual cross-pollination with pollen of a plant donor situated at least 1 m apart. After performing the treatments, each bud was kept isolated from further pollination with the mesh bag, and the flowers were left to develop until fruiting. For each species, we quantified pollination success as in [21], using seed-set per plant to calculate an average self-incompatibility index (ISI), where the seed-set values are the average number of seeds per fruit per treatment (ISI = G/X). ISI index ranges from 0 to 1, where species with values close to zero are considered self-incompatible and those with indices close to one are considered more self-compatible. Then, according to [22], species where at most 20% of the seeds could be produced by self-pollination (ISI ≤ 0.2) were considered self-incompatible (SI), while the remainder were considered self-compatible (SC).

### 2.2. Floral Traits Measurements

Three flowers per plant (*n* = 30–50 plants per population) in the nine *Calceolaria* species were chosen for floral trait measurements. Each flower was photographed from a frontal view with a camera Panasonic Model DMC-LZ20 and three floral traits were measured from pictures using ImageJ 1.46r (http://rsb.info.nih.gov/ij/, Figure 1). We chose floral traits based on their importance for pollinator attraction (corolla area CA; elaiophore area EA) and the effect on the mechanical ability of the plant to self-pollinate (hercogamy index H) as previously documented for ten different *Calceolaria* species and their pollinators [23]. In addition, since preliminary analyses revealed that CA and EA values were significantly correlated (Pearson, r = 0.51, *p* < 0.001), we included the EA/CA ratio as an additional trait that accounts for the total area occupied by the elaiophore gland in the corolla. To test for potential differences in floral morphology between species with different ISIs, we adjusted a Generalized Linear Model (GLM) using a Gaussian distribution.

Finally, to determine if there is a relationship between plant morphology and their reproductive strategies, we used a Principal Components approach, where we evaluated the clustering of plant species in their morphological multidimensional space. In the same analysis, we also evaluated the association between all floral traits and the first two principal components. Here, our expectation was that ISI was positively and significantly explained by morphology. All statistical analyses were performed on standardized log-transformed data in R Studio software version 1.1.453 [24].

### 2.3. DNA Extraction and Genotyping

Three leaves per plant were collected and preserved in silica gel directly in the field (3 leaves × 10 individuals × 9 species = 270 collected samples). DNA extractions were done on ~20 mg of dry material using a modified cetyltrimethylammonium bromide (CTAB) protocol [25]. In order to identify the polymorphic primers for the study species, three leaves per plant were used and the DNA tested with 15 already-published microsatellites [26]. The PCR mix (10 μL) was composed of: 10× PCR buffer, 5 mM MgCl_2_, 2.5 mM dNTP (Invitrogen, Carlsbad, CA, USA), 5 mM forward primer, 5 mM reverse primer, 5 mM fluorescently labeled M13 universal primer, 1U GoTaq G2 (Invitrogen, Thermo Fisher Scientific, Waltham, MA, USA), 1 μL BSA, 20 ng/uL template DNA and H_2_O. PCR cycling conditions were set as follows: 5 min of denaturation at 95 °C, followed by 30 cycles of 1-min at 95 °C, 1-min of annealing at 58 °C, 1-min extension at 72 °C and 10 min of final extension at 72 °C. Then, only those primers that showed amplification (9/15; see Results) were genotyped on ten individuals per species and used to estimate the four genetic diversity parameters (9 primers × 10 individuals × 9 species = 810 samples). PCR products were genotyped in a 3130xl Genetic Analyzer (Applied Biosystems, Life Technologies, ThermoFisher Scientific, Waltham, MA, USA) at the Pontificia Universidad Católica de Chile.

We analyzed all genotypes using GeneMapper v.5 (Applied Biosystems, Foster City, CA, USA). We checked for null alleles using Microchecker v.2.2.3 [27]. We tested for departures from the Hardy–Weinberg equilibrium (HWE) using GenAlEx 6.5 [28]. We quantified genetic variation using several genetic measures: the number of alleles per locus (*Na*), observed (*Ho*) and expected (*He*) heterozygosity, and the fixation index *F_IS_*. Here, our expectation was that SC species would display significantly lower genetic diversity than SI species. In order to determine statistical differences in the genetic diversity parameters between SI and SC species, we ran a Kruskal–Wallis test in R studio [24] between values for the two groups. Additionally, to explore potential phylogenetic signal in the relationship between floral traits and mating system, we performed a Neighbor-Joining (NJ) analysis. For this, we ran 1000 bootstraps on the allelic matrix, then calculated a genetic distance matrix for each bootstrapped dataset, and finally estimated a NJ tree with bootstrap support for the whole dataset. All this was done in DARwin [29].

## 3. Results

### 3.1. Breeding Systems

Our results show that five of the nine *Calceolaria* species tested were unable to develop seeds by automatic self-fertilization, while the four remaining produced just a few seeds per fruit when no pollen vectors were available (Table 1). In addition, our ISI results indicated that four *Calceolaria* species were self-incompatible (SI) and five self-compatible (SC) (Table 1). Specifically, *C. purpurea* and *C. polifolia* were the species with the lowest ISI index (0.01 and 0.05), showing that the species are able to produce only a very small number of seeds by geitonogamy. On the contrary, *C. petiolaris* and *C. lanigera* were the species with the highest ISI index (0.91 and 1), with both species able to develop a high amount of seeds by manual self-pollination (Table 1).

### 3.2. ISI and Morphological Traits

Our results indicate that on average SC species have significantly larger corolla and elaiophore areas than SI species (GLM CA: Estimate = −0.14, t-value = −6.3, *p* < 0.001; GLM EA: Estimate = −0.11, t-value = −2.8, *p* < 0.01), whereas the hercogamy index values and the EA/CA ratio values were not significantly different among groups (GLM, H: Estimate = −0.02, t-value = −0.57, *p* = 0.57; EA/CA: Estimate = −0.01, t-value = −0.82, *p* = 0.41). Specifically, our results show that among SC species, *C. lanigera* displayed the largest corolla area, while *C. filicaulis* ssp. *luxurians* presented the largest elaiophore area and EA/CA ratio, but the lowest hercogamy index (Table 1). *Calceolaria petiolaris* exhibited the smallest corolla area and *C. arachnoidea* the largest hercogamy index and the smallest elaiophore (Table 1). In addition, *C. arachnoidea* and *C. integrifolia* showed the lowest EA/CA ratio. In the SI group, *C. segethii* had the largest corolla area and *C. filicaulis* ssp. *filicaulis* presented the largest hercogamy index, elaiophore area and EA/CA ratio (Table 1). Finally, *C. polifolia* displayed the smallest corolla and elaiophore size, *C. purpurea* had the lowest hercogamy index, and both presented the lowest EA/CA ratio (Table 1).

The Principal Component Analysis (PCA) showed that the two first axes explained 87.68% of the variance (PC1 explained 62.98% and PC2 24.69%; Figure 2). The PCA indicated that the corolla area (CA), elaiophore area (EA) and elaiophore/corolla ratio (EA/CA) of the different species varied independently of the hercogamy index (H). Indeed, CA, EA and EA/CA loaded strongly on PC1, while H loaded strongly on PC2. In the same way, the loading factor (LF) analysis revealed that EA (LF = 0.83) mainly explained the variance of the first component. The second component was almost completely explained by H (LF = 0.91). Our PCA indicated that SC and SI species did not cluster clearly, with many of the species in the two groups overlapping in the multidimensional space (Figure 2a). However, most SC species had large corolla, elaiophore areas and EA/CA ratio (e.g., *C. lanigera* and *C. filicaulis* spp. *luxurians*), while most SI species had smaller corolla size and EA/CA ratio (e.g., *C. purpurea* and *C. polifolia;*
Figure 2b).

### 3.3. Genetic Diversity

Eleven of the fifteen primers tested amplified in most of the plant samples, but only nine were polymorphic in all studied species (Appendix A). However, because some samples had missing alleles, we performed the analysis both with and without a missing-allele filter. Because both methods produced similar results, we present the results obtained with the missing-allele filter, which involved removing any individual with more than 80% missing data (see Appendix A for results using the non-missing-allele filter). The Neighbor-Joining analysis (NJ) identified two main groups, with a different clustering pattern (Figure 3). The first one was a series of well-supported clades formed by all the individuals of three *Calceolaria* species with a self-compatible breeding system (SC). Specifically, this group included all specimens of *C. petiolaris*, *C. filicaulis* spp. *luxurians,* and *C. lanigera.* The second group was little-supported and formed by the remaining six species, both with SC and SI breeding systems (Figure 3). In this group, the only strongly-supported species were *C. segethii* and *C. integrifolia*, while the remainder of the species were placed in a polytomy (Figure 3).

Our allelic analysis indicated that the allele number (N_a_) and observed heterozygosity (Ho) were significantly higher in the SC than in the SI group (Kruskal–Wallis test, *Na*: *X*^2^ = 4.17, df = 1, *p* = 0.04; *Ho*: *X*^2^ = 3.8, df = 1, *p* = 0.05). This was, however, not the case for the remainder of the genetic diversity parameters (*He*: *X*^2^ = 1.24, df = 1, *p* = 0.27; *Fis*: *X*^2^ = 01.98, df = 1, *p* = 0.16). Within the SC species group, *C. filicaulis* spp. *luxurians* showed the highest allele number, while *C. lanigera* exhibited the highest homozygosity, but one of the lowest Fis values (Table 2). On the contrary, *C. petiolaris* was the species with the lowest allele number, observed and expected heterozygosity, and the largest Fis values of the group (Table 2). For SI species, *C. polifolia* was the one with the highest number of alleles per locus, observed and expected heterozygosity values, but the lowest inbreeding index (Fis). The lowest allele number, and expected heterozygosity were estimated for *C. segethii* (Table 2). Finally, the highest Fis value of the group was observed for *C. filicaulis* spp. *filicaulis* (Table 2).

## 4. Discussion

Overall, our self-incompatibility tests indicated that four of the nine species studied here (44%) were self-incompatible. In relation to floral traits, we found that self-incompatible species did not show a reduction in the size of their floral traits. Finally, contrary to our expectations, the microsatellite analysis identified a significant increase in allele number and observed heterozygosity in the self-compatible *Calceolaria* species, while inbreeding parameters were the not significantly different between reproductive groups.

### 4.1. Self-Compatibility and Floral Characters

Floral characters that are directly associated with pollinator attraction and pollen transfer are assumed to be under strong selection by pollination agents (see review in [30]). This is expected to be even more strongly so in specialized pollination interactions, where the pollinator diversity is relatively low, and thus plant reproduction relies only on a reduced number of potential pollinators. In such cases, it could be expected that fluctuations in the abundance or lack of pollinators could lead to the evolution of self-compatibility as a way to assure reproductive success. Our results showed that the studied plants can use different reproductive strategies, with almost half of the evaluated species able to produce an equivalent amount of seeds by selfing and outcrossing, and others identified as strict outcrossers (Table 1).

Even though floral specialization is expected to promote high levels of outcrossing, many plant species with highly specialized pollination systems have the capacity for autonomous selfing [12,13,14]. Therefore, and considering the level of pollinator specialization in the study system [19,23,31], it is not surprising that some of the species studied here are able to self-pollinate at some level. In the present study, the vast majority of the species did not develop seeds by spontaneous self-fertilization, but by geitonogamy (i.e., with the assistance of pollen vectors). These results would suggest that the estimated self-compatibility in some of these species could be related to the development of a strategy that allows the evolution of reproductive assurance mechanisms. Further, recent studies in a few *Calceolaria* species have already revealed their capacity for geitonogamy [18,20,32]. Unfortunately, our experiments did not allow us to identify the timing of selfing (i.e., early or delayed). This is a point that should be further investigated in future works, because it can provide important information on the evolution of the plant-pollinator interaction in the system, and on the (in)ability of pollinators to affect the plant′s fitness, and thus its response to specific selective agents (e.g., pollinator identity, presence/absence of pollinators).

Because a relationship between self-compatibility and pollinator attraction has been proposed and then identified in some systems (see [11]), our expectation was that floral traits associated with attraction and pollen transfer were also going to be affected by the transition from outcrossing to self-pollination, and that this relationship was going to involve reduction of the traits with increases in the ability to self. Even though our results partially support the idea that breeding systems are explained by some floral traits, they do not fully match our initial expectations. On the one hand, we could not observe a clear morphological clustering of SC and SI species, since species with either SI or SC strategies overlap in the multivariate morphological space. Recent investigations in several *Calceolaria* species have documented a morphological relationship between corolla traits, and more importantly, it has been revealed that species with a small elaiophore tend to lose the specialized relationship with their oil-recollecting bees [16,19,23], possibly favoring the appearance of selfing mechanisms. Unfortunately, neither here nor in any past investigation in the genus has it been possible to assess the influence of evolutionary relatedness on the evolution of breeding systems. On this, our NJ analyses could recover some groups that matched species breeding strategies. Our results tend to support that SC species are more phylogenetically closer to each other than with SI species, which might suggest a phylogenetic pattern in the relationship between floral traits and mating strategies. Since this analysis was done based on nine microsatellites and one population per species, future investigations should incorporate more markers and a more extensive phylogenetic approach.

On the other hand, we could not demonstrate that SC species were associated with a reduction in floral traits. Although this result goes against our expectations, it can be explained by the different reproductive strategies that plants can set forth to assure reproductive success in variable pollination environments. It is known that the degree of unreliability of the pollinator environment can lead to different modes of autonomous selfing [15], which seems to be especially important in highly specialized pollination systems where mating events are scarce [14]. In such cases, it would be expected for plants to evolve showier flowers that may be more visible and attractive to the few pollinators that may be available in the environment. This would also agree with some studies showing that delayed selfers have higher hercogamy index and may not experience a reduction of flower size or display [33,34]. Even though our experimental design is appropriate for characterizing the overall reproductive strategy of the species, it does not allow us to determine the timing of selfing. On this, preliminary tests indicate that delayed selfing could be present in at least some *Calceolaria* species, and this should be the focus of future research in the group.

### 4.2. Is Selfing Associated with Lower Genetic Diversity in the Genus Calceolaria?

The transition from outcrossing to selfing has happened several times during the evolution of angiosperms. Further, it has been shown that under certain circumstances, selfing leads to reduced genetic diversity in populations, especially when selfing is instantaneous and early, and thus competes with outcrossing [35]. In this work we evaluated the genetic diversity of nine plant species of the genus *Calceolaria* and asked whether their genetic diversity was associated with different levels of self-incompatibility. In general, we did not find a reduction in genetic diversity or increases in inbreeding in self-compatible species. Indeed, we found that selfers have higher genetic diversity than outcrossers, at least for some indicators (allele number and observed heterozygosity). This result can be explained in different ways. First, it has been shown that under certain conditions (i.e., low inbreeding depression), selfing can evolve in a population and can invade it, becoming an Evolutionary Stable Strategy (ESS) [36,37]. In such a situation, selfers are expected to display low inbreeding depression because those conditions would allow such a strategy to evolve and persist. In this study we show that plant species were unable to develop seeds by automatic selfing, but show the ability to produce seeds by geitonogamy, suggesting the presence of a delayed selfing mechanism. Consequently, it is possible that the maintenance of this mechanism let to SC species to sustain similar (e.g., He and Fis) or even higher genetic diversity values (e.g., Na and Ho) than SI species, since delayed selfing allows maintaining the genetic benefits of a potential outcrossing. *Calceolaria* is a highly specialized plant genus that depends on a few solitary oil-bees for reproduction, so the development of mixed mating strategies rises as an opportunity to affront unreliable pollinator environments. Second, it is important to note that we studied different species collected at only one locality per species. It is thus possible that the lack of signal on some genetic diversity parameters (i.e., He and Fis) is due to the fact that the reproductive strategy is stable at the level we are testing. As we mentioned before, it is possible that in those populations a higher seed production by geitonogamy could have buffered the negative consequences of inbreeding, what would have been reflected in similar Fis values for both groups. Third, in the present study we only found nine polymorphic microsatellites out of the 15 initially tested. It is possible that this low number of markers was not sufficient to capture genetic variability, or that the selected ones were not appropriate to account for differences between the groups under study. For this to be tested, future work should seek to increase the number of samples studied at the intra-specific level (e.g., several populations per species), as well as the number of molecular markers.

## 5. Conclusions

To conclude, we did not observe a reduction of floral attraction traits in self-compatible *Calceolaria* species, but rather found larger corolla and elaiophore areas in this group than in SC species. Also, we did not find lower genetic diversity or higher inbreeding in SC species than in SI species. Even though this goes against our initial expectations, it agrees with previous works. Indeed, it has been observed that plant species involved in specialized pollination interactions maintain mixed breeding systems to affront the unsteadiness of the pollinator environment. In the case of *Calceolaria*, their high dependence on only two genera of oil-bees can put the species in a vulnerable reproductive position, probably facilitating the evolution of mechanisms of reproductive assurance in the absence of pollinators. As a result, the plants maintain their attraction traits while evolving their ability to self. In the context of strong anthropogenic environmental change, this trait in the *Calceolaria* species we investigated may assist species survival, since it would allow the species to adapt to rapidly varying biotic environments.

In species displaying different reproductive strategies, we detected a significant change in the allele number and expected heterozygosity, but not in inbreeding. This suggests that selfing could be delayed, facilitating the exchange of genes by early cross-pollination, and buffering the negative genetic effects of self-pollination. Future studies must further investigate this, ideally in a phylogenetic framework, allowing to finally disentangle the main drivers of breeding system evolution in *Calceolaria*.

## Figures and Tables

**Figure 1 plants-09-01377-f001:**
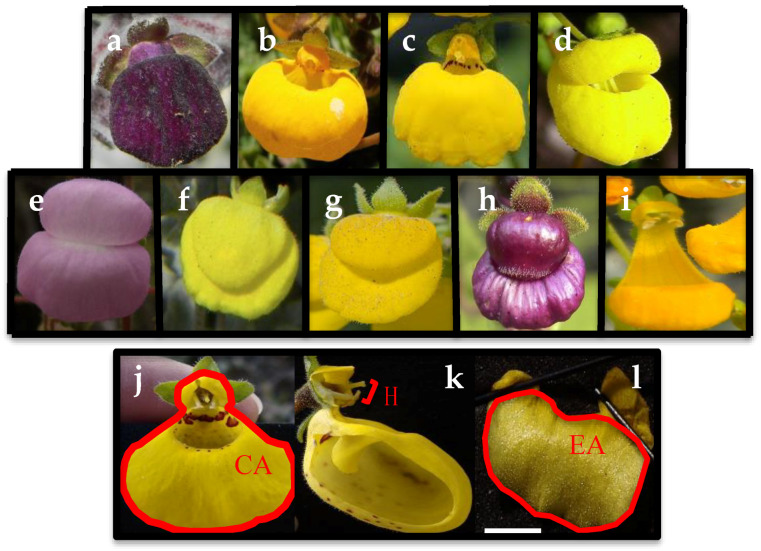
*Calceolaria* species under study and morphological traits measured. (**a**–**i**) plant species: (**a**) *C. arachnoidea*, (**b**) *C. filicaulis* spp. *luxurians,* (**c**) *C. filicaulis* spp. *filicaulis*, (**d**) *C. integrifolia*, (**e**) *C. lanigera,* (**f**) *C. petiolaris,* (**g**) *C. polifolia,* (**h**) *C. purpurea,* and (**i**) *C. segethii*. (**j**–**l**) Measured floral traits depicted in red: (**j**) corolla area CA, (**k**) hercogamy index H, (**l**) elaiophore area EA.

**Figure 2 plants-09-01377-f002:**
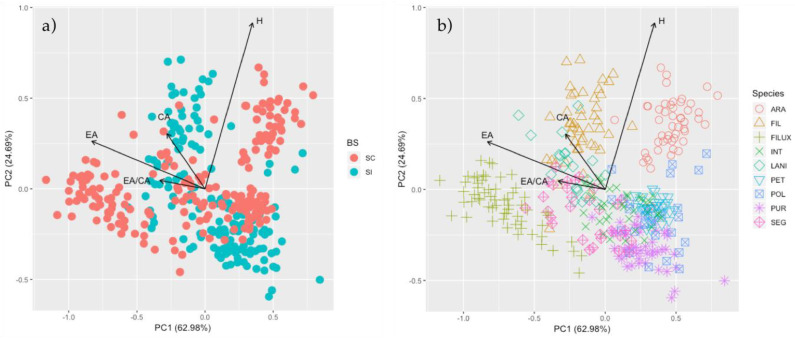
Principal Components Analysis (PCA) of the four floral traits measured in *Calceolaria*. PCA colored by (**a**) breeding system and (**b**) species. PCA vectors for the first and second PCA axes are shown, along with the percentage of explained variance. CA: corolla area, H: hercogamy index, EA: elaiophore area and, EA/CA: elaiophore/corolla area ratio. Species abbreviations are those in Table 1.

**Figure 3 plants-09-01377-f003:**
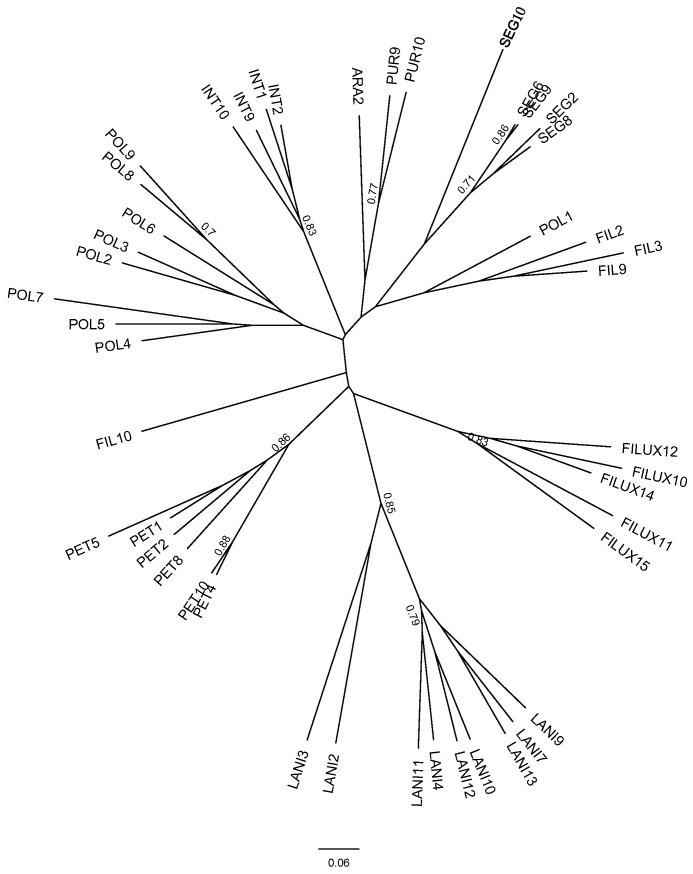
Mid-rooted Neighbor-Joining (NJ) tree of the nine *Calceolaria* species based on distances of nine microsatellite markers. *Calceolaria* species: ARA: *C. arachnoidea*, FIL: *C. filicaulis* spp. *filicaulis*, FILUX: *C. filicaulis* spp. *luxurians*, INT: *C. integrifolia*, LAN: *C. lanigera*, PET: *C. petiolaris*, POL: *C. polifolia*, PUR: *C. purpurea*, and SEG: *C. segethii*. Scale bar indicates genetic distance. Numbers after species name indicate sample names. Bootstrap support values higher or equal to 0.7 are shown on nodes.

**Table 1 plants-09-01377-t001:** Breeding system classification, floral traits measurements, and self-incompatibility indices for the nine studied *Calceolaria* species. BS: breeding system; SI: self-incompatible, SC: self-compatible. PS*: population size indicated as a visual estimation of the number of individuals per population, *n* = sample size. Floral traits: CA: corolla area, H: hercogamy index and EA: elaiophore area. NF: number of flowers used for floral measurements. Reproductive traits: AS: number of seeds per fruit by automatic self-fertilization, X: number of seeds per fruit by cross-pollination and, ISI: self-incompatibility index. Values are given as mean ± standard error. Species abbreviations are shown in parenthesis.

						Morphological Traits				Reproductive Traits (n° Seeds Per Fruit)		
BS	Species	Coordinates	PS*	N	NF	CA (mm^2^)	GA (mm^2^)	EA/CA (mm^2^)	H (mm)	X	AS	ISI
SI	*C. filicaulis* spp. *filicaulis*	36°36′ S/72°00′ W	<100	50	150	100.38 ± 3.52	24.63 ± 1.00	0.25 ± 0.01	5.0 ± 0.37	305 ± 29	0	0.14
	*C. purpurea*	33°23′ S/70°27′ W	0–50	47	141	60.61 ±3.32	6.54 ± 0.34	0.12 ± 0.01	0.85 ± 0.05	90 ± 28	0	0.01
	*C. polifolia*	33°00′ S/70°56′ W	50–100	40	120	57.35 ± 3.99	5.66 ± 0.38	0.12 ± 0.01	2.04 ± 0.21	259 ± 7	0	0.05
	*C. seguetii*	33°21′ S/70°19′ W	50–100	30	90	117.98 ± 8.07	17.28 ± 1.55	0.17 ± 0.02	0.91 ± 0.08	142 ± 40	5 ± 1	0.13
	Mean ± SE					84.08 ± 14.94	13.53 ± 4.55	0.17 ± 0.01	2.22 ±0.97	199 ± 26	1.3 ± 0.25	0.08 ± 0.03
SC	*C. arachnoidea*	36°36′ S/72°00′ W	<100	50	150	94.70 ± 3.38	6.30 ± 0.24	0.07 ± 0.02	8.17 ± 0.47	347 ± 57	0	0.22
	*C. lanigera*	34°14′ S/70°27′ W	0–50	40	120	220.09 ± 18.02	22.08 ± 1.16	0.12 ± 0.01	1.42 ± 0.13	251 ± 62	3 ± 1	1
	*C. petiolaris*	33°23′ S/70°31′ W	0–50	31	93	53.97 ± 1.68	6.42 ± 0.16	0.12 ± 0.004	2.21 ± 0.06	230 ± 35	4 ± 1	0.91
	*C. integrifolia*	33°22′ S/70°24′ W	0–50	37	111	142.62 ± 8.73	8.86 ± 0.56	0.07 ± 0.001	1.24 ± 0.07	231 ± 73	3 ± 2	0.23
	*C. filicaulis* spp. *luxurians*	33°19′ S/70°16′ W	<100	55	165	138.56 ± 7.62	58.83 ± 2.94	0.47 ± 0.03	0.39 ± 0.03	224 ± 121	0	0.61
	Mean ± SE					129.99 ± 27.73	20.50 ± 10.02	0.17 ± 0.02	2.69 ± 1.40	257 ± 70	2 ± 0.8	0.53 ± 0.16

Note: Asterisk denotes a variable for which the values were not estimated in the present study, but extracted from unpublished data (Murúa, Pers. Obs).

**Table 2 plants-09-01377-t002:** Parameters of genetic diversity estimated for all *Calceolaria* species using nine microsatellites. BS: breeding system, SI: self-incompatible, SC: self-compatible. Na: allelic number, Ho: observed heterozygosity, He: expected heterozygosity and F_IS_: inbreeding index. Values are given as mean ± standard error.

BS	Species	Na	Ho	He	F_IS_
SI	*C. filicaulis* spp. *filicaulis*	3.00 ± 0.5	0.18 ± 0.08	0.35 ± 0.08	0.63 ± 0.20
	*C. purpurea*	3.44 ± 0.5	0.19 ± 0.09	0.51 ± 0.07	0.53 ± 0.20
	*C. polifolia*	3.78 ± 0.3	0.31 ± 0.08	0.55 ± 0.06	0.41 ± 0.14
	*C. segethii*	2.56 ± 0.4	0.08 ± 0.06	0.30 ± 0.08	0.71 ± 0.14
	Mean ± SE	3.20 ± 0.27	0.19 ± 0.04	0.43 ± 0.06	0.57 ± 0.06
SC	*C. arachnoidea*	3.78 ± 0.8	0.34 ± 0.13	0.46 ± 0.09	0.38 ± 0.24
	*C. lanigera*	5.00 ± 0.6	0.34 ± 0.09	0.57 ± 0.03	0.35 ± 0.18
	*C. petiolaris*	2.56 ± 0.4	0.13 ± 0.06	0.35 ± 0.08	0.56 ± 0.17
	*C. integrifolia*	2.89 ± 0.4	0.20 ± 0.05	0.36 ± 0.08	0.35 ± 0.12
	*C. filicaulis* spp. *luxurians*	6.67 ± 0.9	0.32 ± 0.08	0.65 ± 0.06	0.51 ± 0.09
	Mean ± SE	4.18 ± 0.75	0.27 ± 0.04	0.48 ± 0.06	0.43 ± 0.04

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
