# Peer review of "Self-Compatibility Not Associated with Morphological or Genetic Diversity Reduction in Oil-Rewarding Calceolaria Species"

_plants, 2020, doi:10.3390/plants9101377_

Round 1

Reviewer 1 Report

In the manuscript authors discussed intended to elucidate relation between floral biology and genetics, studying self-compatibility, floral traits, and genetic diversity.

Authors raised following 3 objectives,

a) test breeding system of target 9 species

b) demonstrate reduction in the flora traits associated with pollinator interaction

c) assess self-compatibleness and genetic diversity

Although the results did not support all the authors hypotheses, approach of the study sounds interesting and the results will be significant to elucidate establishment mechanism of self-compatibility.

Comments on the manuscript

Number of individuals per population tested for genetic diversity evaluation need to be increased. The numbers of individuals tested to evaluate genetic diversity, 3 individuals per species collected from one population, seems insufficient to assess self-compatibleness and genetic diversity. Authors discuss relations between genetic diversity values and self-compatibleness. Delayed selfing assumption sounds interesting and could be its reason of higher diversity value of SC than SI, but small number of individual tested remains as reason why discussion part is not much persuasive (authors themselves says that “we are here comparing data obtained from different species and that we collected at one locality per species only.”) as authors themselves described in “3.2 Is selfing associated with lower genetic diversity in the genus Calceolaria?”.

Population size are shown in Table 1, but no information provided distribution of populations in each locality. Authors could discuss about genetic diversity values from geneflow or population isolation point of view, if information of population distribution are provided (e.g. only one isolated population, or many small populations were found surrounded area, or one big population etc.).

Specific and technical comments

1) Page 2, Figure 1 (b) No image available.

2) Page 4, Figure 3, Figure legend is not distinguishable. Please combine different colors and shape of symbols.

3)Page 5, Figure 4, only few names of the individuals are shown. It is recommended to make the tree larger so that the names of individuals (or at least abbreviation of the species) are visible.

4) Page 8, line 4-5, “the observed heterozygosity (Ho) were significantly different”, did authors tested it using any statistical method? If so, please add the description of the method.

5) Page 8, 4. Materials and methods, 2nd paragraph, “…different regions throughout central Chile (Figure 1)” does not coincide with the Figure on page 2.

6) Page 9, 4.3 DNA extraction and genotyping, “Fifteen microsatellites previously described by [34]” were tested …” Marker information is not sufficient and marker names do not coincide in the way described in [34].

7) Page 9, 4.3 DNA extraction and genotyping, no validation of phylogenetic trees. Authors should assess the variability in the obtained topology.

Author Response

Dear Reviewer 1:

Accompanying this letter, you will find a new version of our manuscript entitled “Self-compatibility not associated with morphological or genetic diversity reduction in oil-rewarding Calceolaria species”, authored by Maureen Murúa, Anahí Espíndola and Fernanda Pérez.

Here, you will find a new revised version of the manuscript, where we address the all your comments. We hope that this submission will satisfy your requirements and will be considered for publication in Plants.

Best regards,

Dr. Maureen Murúa

Universidad Mayor

Chile

Answers to the comments are given in italics.

Comment 1:

Number of individuals per population tested for genetic diversity evaluation need to be increased. The numbers of individuals tested to evaluate genetic diversity, 3 individuals per species collected from one population, seems insufficient to assess self-compatibleness and genetic diversity. Authors discuss relations between genetic diversity values and self-compatibleness. Delayed selfing assumption sounds interesting and could be its reason of higher diversity value of SC than SI, but small number of individual tested remains as reason why discussion part is not much persuasive (authors themselves says that “we are here comparing data obtained from different species and that we collected at one locality per species only.”) as authors themselves described in “3.2 Is selfing associated with lower genetic diversity in the genus Calceolaria?”.

A: We apologize to the reviewer for this misunderstanding. The sampling of 3 individuals was only used for testing the primers we were planning to use in more samples later on in our work. Once the polymorphic primers were identified (in this case, 9 out of the 15 tested), they were genotyped in 10 individuals per species, not 3. To clarify this, we have rephrased this throughout section 2.3 (see page 7, lines 151-155).

Regarding the arguments given in the discussion section, they were raised in a conservative way thinking that these analyzes were based only on one population per species, not on the number of individuals per species (10 individuals). Therefore, according to this comment and those raised by the other reviewers we have now improved section 2.3 (see page 7, lines 141-166).

Comment 2:

Population size are shown in Table 1, but no information provided distribution of populations in each locality. Authors could discuss about genetic diversity values from gene flow or population isolation point of view, if information of population distribution are provided (e.g. only one isolated population, or many small populations were found surrounded area, or one big population etc.).

A: Based on this comment, we have now revised the legend of Table 1, to make clear that this value corresponds to a visual estimation of the population size in the population visited. We would like, however, to mention that we do not feel comfortable stating specific population sizes, since our sampling strategies were not targeting this value, and were rather geared towards providing a general overview of the populations sizes we could observe.

Specific and technical comments

Comment 1:

Page 2, Figure 1 (b) No image available.

A: This was an upload error in the submission. We will make sure that it does not happen again this time around. We request the Editor to let us know if this is the case, and we will re-upload the Figure.

Comment 2:

Page 4, Figure 3, Figure legend is not distinguishable. Please combine different colors and shape of symbols.

A: We appreciate this comment. Based on this, the new version of Figure 3 includes different colors and symbols.

Comment 3:

Page 5, Figure 4, only few names of the individuals are shown. It is recommended to make the tree larger so that the names of individuals (or at least abbreviation of the species) are visible.

A: We thank the reviewer for pointing this. Based on this comment, and using a new revised Figure, we now include a larger figure with all taxon names visible, as suggested (see Figure 3).

Comment 4:

Page 8, line 4-5, “the observed heterozygosity (Ho) were significantly different”, did authors tested it using any statistical method? If so, please add the description of the method.

A: Statistical differences between SC-SI groups were tested using Kruskal-Wallis test, which is mentioned in the last paragraph of the section 2.3 (see page 8, lines 162). The results of those analyses are described in the Results section, specifically in the second paragraph in section 3.3 (see page 12, lines 243-246).

Comment 5:

Page 8, 4. Materials and methods, 2nd paragraph, “…different regions throughout central Chile (Figure 1)” does not coincide with the Figure on page 2.

A: We apologize for this mistake; we should have indeed mentioned Table 1 instead of Figure 1. This has been now corrected in the new version (see page 5, lines 103-107), and we thank the reviewer for pointing this out. 

Comment 6:

Page 9, 4.3 DNA extraction and genotyping, “Fifteen microsatellites previously described by [34]” were tested …” Marker information is not sufficient and marker names do not coincide in the way described in [34].

A: In the cited publication where the microsatellites were described by first time, the loci were renamed for simplicity. In order to make the connections between the two manuscripts clear, and to allow for consistency, we have now added the Genbank accession numbers of each locus (see Table S1, Supplementary Materials).

Comment 7:

Page 9, 4.3 DNA extraction and genotyping, no validation of phylogenetic trees. Authors should assess the variability in the obtained topology.

A: We thank the reviewer for this observation. Taking this into consideration, we have now performed a bootstrapping approach on the dataset, and phylogenetic support has now been calculated on the estimated tree(s). The methods and results for this analysis are now presented throughout the manuscript and in the new Figure 3. We think that this new analysis adds strength to our results, and we are thankful to the reviewers for this recommendation.

Reviewer 2 Report

I found this paper interesting and enjoyed reading it. It was easy to understand the logic behind the hypotheses and the tests was very clearly and carefully explained. The field work is impressive and the data set seems accordingly to be of high quality. I also find the results reasonable, although they might seem to be some unexpected to the authors. I find this manuscript worthy publication.

I have some small extra comments:

I would considered changing the title to Self-Compatibility seems not to be Associated toMorphological or Genetic Diversity Reduction in theOil-Rewarding Calceolaria Species. (due to the relatively small number of polymorphic microsatellites available for testing)

Figure b is missing content in my version

Presentation of Results and Discussion before Material and Methods (in my version) seems not to be logic

An another interpretation of the PCA is that Herkogami variation is independent of corolla area (CA) and elaiophore area (EA) and visa versa

The first four lines of the discussion is repetition and might be deleted if shortage of space in the journal

Would it be wise to possibly address and give answer to these questions? Do you know approximately how many species belonging to the two pollinator genera’s? Are there any information on changes in the total population of the pollinators on a regional scale?

Is it possible to make any general plant ecology statements from your results? It would have been interesting to hear if your opinion on how your results could contribute in the debate on current global biodiversity decline due to such as climate change, increased human developments and land use, pollution or invasive species.

Author Response

Dear Reviewer 2:

Accompanying this letter, you will find a new version of our manuscript entitled “Self-compatibility not associated with morphological or genetic diversity reduction in oil-rewarding Calceolaria species”, authored by Maureen Murúa, Anahí Espíndola and Fernanda Pérez.

Here, you will find a new revised version of the manuscript, where we address the all your comments. We hope that this submission will satisfy your requirements and will be considered for publication in Plants.

Best regards,

Dr. Maureen Murúa

Universidad Mayor

Chile

Answers to the comments are given in italics.

I found this paper interesting and enjoyed reading it. It was easy to understand the logic behind the hypotheses and the tests was very clearly and carefully explained. The fieldwork is impressive and the data set seems accordingly to be of high quality. I also find the results reasonable, although they might seem to be some unexpected to the authors. I find this manuscript worthy publication. I have some small extra comments.

Comment 1:

I would considered changing the title to Self-Compatibility seems not to be Associated to Morphological or Genetic Diversity Reduction in the Oil-Rewarding Calceolaria Species. (due to the relatively small number of polymorphic microsatellites available for testing)

A: Based on this comment, we have now revised the title, which is now “Self-compatibility not associated with morphological or genetic diversity reduction in oil-rewarding Calceolaria species”. Even though not exactly what the reviewer suggested, we think that it is both more concise and carries the same message as that suggested by the reviewer. We appreciate this comment and we agree that this title seems to be more appropriate considering our results.

Comment 2:

Figure b is missing content in my version

 A: We apologize for this mistake; there was a problem of the downloaded version. We provided an individualized version of the images in pdf format in the initial submission, but we do not know if you had access to them. In order to prevent this from happening again, the new version has all figures and legends included directly in the text.

Comment 3:

Presentation of Results and Discussion before Material and Methods (in my version) seems not to be logic

A: We apologize for this. Indeed, this special formatting was due to our understanding of the journal’s formatting. Taking this into consideration, we have now modified the new version so as to carry the ‘normal’ formatting.

Comment 4:

Another interpretation of the PCA is that Herkogami variation is independent of corolla area (CA) and elaiophore area (EA) and visa versa

 A: Thank you so much for pointing this out. Based on this, we now include this observation in the Results section (see page 10, lines 205-208).

Comment 5:

The first four lines of the discussion is repetition and might be deleted if shortage of space in the journal

A: As suggested, this part was removed from the discussion section (see page 14, Discussion section).

Comment 6:

Would it be wise to possibly address and give answer to these questions?.

1. Do you know approximately how many species belonging to the two-pollinator genera’s?

 A: According to Neef et al. 2017 Centris has approximately 200 species and Chalepogenus about 21 described species. This is of course an estimation, since recent published and unpublished studies seem to suggest that there is more diversity than what is currently considered. Even though we think this is interesting, we feel that this information would be distracting if added to the manuscript, and have decided for that reason to not include it in the current version. 

Neef, J. L.; Simpson, B.B. Vogel’s great legacy: The oil flower and oil-collecting bee syndrome. Flora, 2017, 232, 104-116.

2. Are there any information on changes in the total population of the pollinators on a regional scale?

A: To our knowledge there is no information about population dynamics of any of the pollinator species and at any scale. This is an extremely poorly understood topic, and is something that some groups are currently studying. Because of this lack of information, this has not been included in the current version of the manuscript.

3. Is it possible to make any general plant ecology statements from your results?

A: We thank the reviewer for this comment, but we are a bit confused about what the reviewer is referring to in this comment. Indeed, our manuscript includes quite some information about the plant ecology, referring to studies previously done on the pollination ecology and distribution of the different species.

4. It would have been interesting to hear if your opinion on how your results could contribute in the debate on current global biodiversity decline due to such as climate change, increased human developments and land use, pollution or invasive species.

A: Based on this comment, we have now added in the Conclusion a short mention to this topic (see page 17). We thank the reviewer for this recommendation.

Reviewer 3 Report

The work is dedicated to the intriguing topic and uses a very interesting model, Calceolaria, which is poorly investigated to date. That is why the outcome of this survey may be of significant interest to readers.

However, this outcome seems somewhat modest and needs elaboration to be more reliable. Data in Table 1 are presented in the way which needs adjustment. For some traits they should be given as average ± S.E. Moreover, there should be results of statistically relevant comparison instead of statements like 'bigger corolla' or 'almost the same'. Surprisingly, there are no estimation of correlation between studied parameters.

Term 'hercogamy' traditionally refers to strategy, state or phenomenon, which cannot be measured, so I suggest to call this trait in some other way.

The discussion in section 3.2. is given as if authors themselves find their own results unreliable or artifact. This section needs either elaboration to highlight what is new in results concerning DNA polymorphism and what they tell us about, or omitting, as in its present form these data are debatable.

There are numerous suggestions and corrections directly into the manuscript, see attachment.

I hope that all these comments and suggestions will serve better understanding the authors' position. After their accepting (or rebuttal against some of them), this paper can be recommended for publication.

Author Response

Dear Reviewer 3:

Accompanying this letter, you will find a new version of our manuscript entitled “Self-compatibility not associated with morphological or genetic diversity reduction in oil-rewarding Calceolaria species”, authored by Maureen Murúa, Anahí Espíndola and Fernanda Pérez.

Here, you will find a new revised version of the manuscript, where we address the all your comments. We hope that this submission will satisfy your requirements and will be considered for publication in Plants.

Best regards,

Dr. Maureen Murúa

Universidad Mayor

Chile

Answers to the comments are given in italics.

Major comments

The work is dedicated to the intriguing topic and uses a very interesting model, Calceolaria, which is poorly investigated to date. That is why the outcome of this survey may be of significant interest to readers. However, this outcome seems somewhat modest and needs elaboration to be more reliable.

Comment 1:

Data in Table 1 are presented in the way, which needs adjustment. For some traits they should be given as average ± S.E. 

A: Based on this comment, we have now corrected Table 1.

Comment 2:

Moreover, there should be results of statistically relevant comparison instead of statements like 'bigger corolla' or 'almost the same'. Surprisingly, there are no estimation of correlation between studied parameters.

A: In our opinion, it is appropriate to provide some descriptive information that accounts of the dimensions of the species traits, rather than just comparisons regarding their averages as groups and their statistical differences. This system has been little studied, especially in Chile, so any biological description is both novel and of interest. Therefore, we request that these references are maintained in the new version of the manuscript. However, because we take the reviewer’s point seriously, we have made sure that these references are accompanied by the corresponding statistical results. Additionally, we noticed that the statistical differences of such descriptions were wrongly positioned in the text (at the end) so we decided to relocate and rephrase them for a better understanding.

            Regarding to the correlation analysis, in the submitted version we did not include any results about the correlation between traits. However, following this suggestion we performed a preliminary correlational analysis and corroborate that CA and EA were correlated, so we have now briefly added this information in the Material and Methods section (see page 6, lines 129-131).

Comment 3: Term 'hercogamy' traditionally refers to strategy, state or phenomenon, which cannot be measured, so I suggest to call this trait in some other way.

A: We thank the reviewer for this comment. In fact, we came to realize that this was due to a change in the way scientists have been approaching the concept over the years. In the submitted version we referred to “Hercogamy” as the continuous value that measures the distance between reproductive structures, which we understood as a valid reproductive trait. However, some recent publications indicate that some authors refer to it as a strategy rather than a measurable trait. To clarify this, and following Lazaro et al. 2020, we have now decided to use the term “hercogamy index” to describe our measures. This term refers to a continuous value that measures the distance between reproductive structures, which corresponds to what we use in our study. This change was implemented in the new version of the manuscript (see page 6, lines 126-129, Materials and Methods section).

Lazaro, A., Seguí, J; Santamaría, L. Continuous variation in hercogamy enhances the reproductive response of Lonicera implexa to spatial variation in pollinator assemblages. AoB Plants, 2020, 12, 1-10.

Comment 4: The discussion in section 3.2. is given as if authors themselves find their own results unreliable or artifact. This section needs either elaboration to highlight what is new in results concerning DNA polymorphism and what they tell us about, or omitting, as in its present form these data are debatable.

A: We thank the reviewer for this comment and based on it we have now taken care of clarifying it. On this, we think that even though a relatively small sampling of microsatellites, our results are not unimportant. That said, we are also aware of the limitations of our approach, and this is what we were trying to present in the discussion of the results. We suspect that this discussion is what the reviewer observed as us considering our results “unreliable”. Because we want to make sure that our analysis stresses both the strengths and weaknesses of our study, we have now revised this section to make this clear. We hope the reviewer and the Editor with find this satisfactory (see page 16, Discussion: section 4.2).

Minor comments

These comments are inserted on the PDF document and they will be answered in the PDF or here when necessary.

Comment 1:

What do authors mean by 'genetics'? Is it anything about genetic control of floral development or changes in genetic structure of population conditioned by changes in pollination mode? Please rephrase to specify.

A: We appreciate this comment, this has been now properly clarified this in the abstract.

Comment 2:

What do authors mean by the 'higher morphological consequences'?

 A: We apologized for this expression. We had actually intended to explain that it is expected that the negative effects of selfing, both morphological and genetic, could be greater for automatic self-fertilized plant species. This line was rephrased to better communicate the idea (see page 3, lines 57-59).

Comment 3:

I do not understand what it means. Please rephrase.

A: In retrospect, we now understand that the phrase was a little confusing. Therefore, we have now rewritten it (see page 3, lines 60-62) and we are confident that its clarity has been now improved. 

Comment 4:

Hercogamy is a strategy, not a structure. How can it be reduced like areas or sizes?.

A: As we mentioned in major revision comment 3, we have now changed the name to “Hercogamy index” (for more detail see comment 3 in major comment section).

Comment 5:

In my opinion, this figure needs reformatting. First, it is not a good idea to insert 'a' and 'b' and then additional 'a' and 'b' within 'a'. Second, as I may see, photo images of flowers (a) do not fit their frames, so these frames need to be made wider. Third, I cannot see the images at 'b' (possibly it is a problem of downloaded version), but, if judged by figure caption, there are no scale bars even in morphological images. Please add them.

A: We appreciated this comment. Unfortunately, this was a problem of the downloaded version. In fact, in the previous version we had also attached the figures in pdf format so that the reviewers could download them individually and review them better (we do not know if the reviewers were finally able to access them). In the new version of the manuscript, the letter nomenclatures have been corrected so as to refer to each species (see page 5, new Figure 1).

Comment 6:

Were these differences and similarities statistically reliable? I cannot see any estimations of significance of differences (such as p-value) neither in Table 1, nor in Fig. 2.

A: We apologize for this error. The information of the statistical differences was provided at the end of this section. Based on your suggestions, we have now edited the section and this has been now rephrased (see page 9-10, lines 192-196). 

Comment 7:

I think that Table 1 and Figure 2 represent the same data. Possibly it is worth removing Fig. 2.

In its given form, this figure is difficult to understand, as there are no comments what box and whiskers correspond to.

A: Thanks for suggesting this. Based on it, we have now decided to remove the previous version of Figure 2.

Comment 8:

Are these features correlated with each other? If yes, this grouping is somewhat trivial. Possibly it is worth inserting additional parameter, such as EA/CA, i.e., how big EA is in comparison with the whole corolla.

A: We agree with this observation since the elaiophore gland and corolla area are in fact correlated. Therefore, we decide to briefly include the correlation values and EA/CA ratio as an additional trait as the reviewer suggest (see Material and Methods and Results sections).

Comment 9:

I cannot see any evaluation of reliability of nodes in this tree (e.g., bootstrap values). They should be presented, at least the ones exceeding 50.

A: We agree with this comment. The new version of the manuscript includes a new figure with support values obtained using a bootstrap analysis (see new Figure 3).

Comment 10:

What does 'fluctuations of pollinators' mean?

A: We apologize for this inaccuracy. We refer to the fluctuations in the pollinator’s abundance. We have now clarified this in the text (see page 14, lines 275).

Comment 11:

I don't like the way how these data are presented. They should either be published with an appropriate reference provided, or incorporated into this article. Otherwise there are many questions about methods of this study which unfortunately remain unexplained anywhere. Even (Murúa, in press) with expected references would be better than (Murúa, unpublished data) without any additional explanations.

A: We agree with this comment. In order to improve the discussion and to improve the focus in our data, we have now decided to remove this information from the manuscript.

Comment 12:

It could mean that your list of features and measurements is not fully exhaustive (nor fully representative) to characterize pollination syndrome in studied species.

A: We apologized for this misunderstanding. It is clear that we are only working with 9 species of the Calceolaria described for the genus, which can result in a bias when seeking to identify general patterns in the group. Even though this is true, we do believe that the results obtained here are mainly explained by the mixed strategies followed by the vast majority of these species, and that this is not necessarily related to trait or species choice. Indeed, this is the first study to investigate this in Calceolaria, and we think it is important that this is published. However, because we think it would be important to make sure that these results are taken at face value, we now present and discuss them in a rather conservative way. Indeed, we are aware that future work on the system will need to increase the number of species and their respective phylogenetic correction to reinforce the trends documented here. This is now mentioned in the manuscript see page 14, Discussion: 4.1 section).

Comment 13:

Actually, I cannot understand why expected that mode of pollination (or, more strictly, three measured traits) must have any phylogenetic significance. Smaller or larger flowers (as well as elaiophores) could have arisen repeatedly in course of Calceolaria evolution.

A: We appreciate this comment. In a past version of the manuscript we did not perform any phylogenetic analysis since we were not expecting or looking for any phylogenetic signal in any of the studied traits. But, when we carefully analyzed the PCA output, we though that the absence of a clear clustering pattern could be a phylogenetic signal, and thus this hypothesis had to be properly tested. This is the reason why we decided to carry out a NJ analysis. Because we think that even though unexpected, the macroevolutionary hypothesis needs to be included, we decide to maintain this analysis in the manuscript. Along with this, we think that this makes it helpful to fully understand the multivariate results. 

Comment 14:

The first paragraph should be either moved to the 'Introduction' or omitted, as it has nothing to do with materials or methods. Again, please check if all Latin names are italicized.

A: Following this suggestion this part was removed from the Material and Methods.

Round 2

Reviewer 1 Report

Authors have response point by point, and the manuscript is acceptable without correction. Presentation of Figures are greatly improved.

I appreciate effort made by authors and I look forward seeing authors’ futures work on identifying the main drivers of breeding system evolution of the target species.

Reviewer 3 Report

Dear colleagues,

I'm glad that you considered my suggestions founded and helpful. Hopefully these may improve the manuscript.

The only think I may additionally recommend (not regarding the discussed paper but possibly for future work) is that the term 'index' suits better ratios rather than direct measurements. In the other words, 'index' is typically unitless, while the 'hercogamy index' you've applied can be represented in cm, mm etc.

It is my pleasure to recommend your paper for publication in 'Plants'.